# Robot Intelligent Grasp of Unknown Objects Based on Multi-Sensor Information

**DOI:** 10.3390/s19071595

**Published:** 2019-04-02

**Authors:** Shan-Qian Ji, Ming-Bao Huang, Han-Pang Huang

**Affiliations:** Robotics Laboratory, Department of Mechanical Engineering, National Taiwan University, Taipei 10617, Taiwan; shanqianji@gmail.com (S.-Q.J.); d00522011@ntu.edu.tw (M.-B.H.)

**Keywords:** contact modelling, force and tactile sensing, grasping and manipulation, grasp planning, object features recognition, robot hand-arm system, robot tactile systems, sensor fusion, slipping detection and avoidance, stiffness measurement

## Abstract

Robots frequently need to work in human environments and handle many different types of objects. There are two problems that make this challenging for robots: human environments are typically cluttered, and the multi-finger robot hand needs to grasp and to lift objects without knowing their mass and damping properties. Therefore, this study combined vision and robot hand real-time grasp control action to achieve reliable and accurate object grasping in a cluttered scene. An efficient online algorithm for collision-free grasping pose generation according to a bounding box is proposed, and the grasp pose will be further checked for grasp quality. Finally, by fusing all available sensor data appropriately, an intelligent real-time grasp system was achieved that is reliable enough to handle various objects with unknown weights, friction, and stiffness. The robots used in this paper are the NTU 21-DOF five-finger robot hand and the NTU 6-DOF robot arm, which are both constructed by our Lab.

## 1. Introduction

Rapid technology development is enabling intelligent robots to be used in many fields, such as medicine, the military, agriculture, and industry. A robot’s ability is a key function to grasp and manipulate an object that helps people with complicated tasks. In order to provide daily support by using humanoid hands and arms [1,2], robots must have the ability to grasp a variety of unseen objects in human environments [3].

In the unstructured environment, a common gripper has the limitation of not being able to grasp a great variety of objects. Therefore, studies need to be focused on using a multi-fingered robot hand to grasp objects with different shapes. Second, recognition and grasping of unknown objects in a cluttered scene have been very challenging to robots. This study attempted to develop a grasping system that is fast, robust and does not need a model of the object beforehand in order to reduce reliance on preprogrammed behaviors. The contributions of this work are summarized as follows:We present a method that combines three-dimensional (3D) vision and robot hand action to achieve reliable and accurate segmentation given unknown cluttered objects on a table.We propose an efficient algorithm for collision-free grasping pose generation; the grasp pose will be further checked for its grasp quality. Experiments show the efficiency and feasibility of our method.We addressed the problem associated with grasping unknown objects and how to set an appropriate grasp force. To fulfill this requirement, we adopted a multi-sensor approach to identify the stiffness of the object and detect slippage, which can promote a more lifelike functionality.By fusing all available sensor data appropriately, an intelligent grasp system was achieved that is reliable and able to handle various objects with unknown weights, friction, and stiffness.

## 2. Preliminary Knowledge

### 2.1. Point Cloud Processing

For a grasping task [4,5,6,7], finding the position and pose of the target object must be done first. The real environment (Figure 1a) is complex, and the raw point cloud is typically dense and noisy. This will hurt performance in object recognition (Figure 1b). Many point cloud processing methods can be used to deal with the point cloud set, and then useful information can be extracted to reconstruct the object.

#### 2.1.1. Pass-Through Filter

The raw data of a point cloud contains numerous useless points. Therefore, a pass-through filter is used to set a range over the 3D space so that points within that range are kept unchanged, and points outside of the range are removed. In Figure 2, we show the point cloud after using a pass-through filter.

#### 2.1.2. Down-Sampling

A point cloud might contain points that provide no additional information due to noise or inaccuracy of the capture. On the other hand, when there are too many points in a point cloud, processing can be computationally expensive. The process of artificially decreasing the number of points in a point cloud is called down-sampling. In this study, the algorithm we used was voxel grid down-sampling. Figure 3 shows a point cloud before down-sampling and the same point cloud after down-sampling.

#### 2.1.3. Random Sample Consensus (RANSAC)

First, we assumed objects are placed on flat surfaces. In a point cloud representation of a scene, a random sample consensus [8] can help to distinguish flat surfaces. Isolating a point cloud cluster that represents the unknown object is done by removing the entire points on the flat surface. After the planar model has been found, we can easily remove the ground and keep those objects on the ground. Figure 4 shows the objects retrieved from the original point cloud after image processing.

#### 2.1.4. Euclidean Segmentation

After separating the object from the table, the next problem was how to detect the number of objects remaining. The simple method we used was Euclidean segmentation. The result is shown in Figure 5, where we used different colors to represent different objects.

### 2.2. Oriented Bounding Box and Decomposition

For a common object, there are many possible grasps, and searching all possible grasps is difficult and unrealistic. In order to select a good grasp for an unknown object, we adopted an oriented bounding box (OBB) to identify the shape and pose of objects. The box is one of the most common shape primitives to represent unknown objects [6,9,10,11], and it can help us quickly find a suitable grasp. We assume that a hand can grasp an object if the bounding box of the object can be grasped. We created the OBB via principal component analysis (PCA). The steps involved in the OBB method are as follows:Compute the centroid (c0, c1, c2) and the normalized covariance.Find the eigenvectors for the covariance matrix of the point cloud (i.e., PCA).These eigenvectors are used to transform the point cloud to the origin point such that the eigenvectors correspond to the principal axes of the space.Compute the max, min, and center of the diagonal.

Figure 6 shows a joystick, a glass case, and a tennis ball, each with their bounding box as generated using the algorithm discussed above.

In order to get better resolution of the actual models, we approximated the objects with bounding box decomposition. For this process, the number of boxes needed depends on the accuracy of approximation that is required. The principle of bounding box decomposition is that each bounding box is six-sided, and the opposing sides are parallel and symmetric. Considering the computation time, we determined that there were three valid split directions that were perpendicular to planes A, B, and C (Figure 7). The goal was to find the best split plane between each of these opposing sides.

The split measure was defined as:(1)α(d, x),
where *d* is one of the three split directions and *x* is the split position on this axis. For each *x* that cuts the target point cloud, the original point cloud can be divided into two subsets of data points. These can be used as inputs for the OBB algorithm to produce two child bounding boxes. Then we can obtain α(d, x) as the fraction between the sum volume of the two parts of the OBB and the whole volume. It is intuitive that the minimum is the best split (Figure 8). In this way, the box and the data points can be iteratively split, and the new boxes will better fit the shape.

Additionally, for the purpose of efficiency, an iterative breaking criterion was set as:(2)$$\Theta^{*}=\frac{V(C_{1})+V(C_{2})}{V(P)}\geq \theta ,$$
where *P* is the current (parent) box, $C_{1}$ and $C_{2}$ are the two child boxes produced by the split, *V* is a volume function, and θ is the threshold value.

The bounding box decomposition algorithm has two constraints. First, if the Θ is too high, the split is not valuable. Second, boxes that include a very low number of points that can be considered noise must be removed.

### 2.3. Object Segmentation in a Cluttered Scene

When there are objects in a cluttered scene, it is difficult for a robot to distinguish among the unknown objects and grasp them separately. However, a robot’s object segmentation ability is key when it must work in human environments. In our study, the goal was to distinguish the object states by using the robot visual system and inference of object logic states through taking specific actions. This can instruct a robot about an accurate segment object point cloud in the scene without supervision.

The framework of unknown object segmentation is shown in Figure 9. First, the vision module was used to capture the 3D point cloud of the current task environment, and the unknown object was segmented through Euclidean segmentation and saved as the original object dataset. At the same time, the current visual detection results were sent to the object logic state inference module to generate the initial logic state of each object. This was done because the spatial position relationship between the objects was unclear, such as with the presence of an occlusion and superposition. The initial object dataset is a generally inaccurate representation of the current environment.

Specifically, when the robot performs a certain action, it feeds back the execution result of the action to the object logic state inference module, updates the logic state of the related object, and then updates the object point cloud. By observing the point clouds of objects that do not conform to the changed logic state, the object that is blocked or superimposed can be found, and the visual inspection result is fed to the object logic state inference module to update the object’s logic state space.

First, define the object and logic state:(3)$$s_{i}=[Z_{1},\;Z_{2}{,\;Z}_{3}],$$
where $Z_{1}$ is equal to 1 or 0, which indicates if object *i* is in or out of the robot’s hand. Variable $Z_{2}$ presents the number of objects that are piled around object *i*. If $Z_{2}$ is equal to 0, it indicates that no objects are piled around object *i*. Variable $Z_{3}$ represents the number of objects under object *i*. If $Z_{3}$ is equal to −1, it indicates that object *i* is in the air. If variable $Z_{3}$ is equal to 0, it indicates that object *i* is on the ground.

According to the definition of object logic state, the two most important robot actions must be the focus: PickUp() and PutDown(). The action PickUp(object *i* ) can be used to grasp object *i* from the table, and PutDown(object *i*, *j* ) can be used to load the grasped object *i* into the target position *j*. It is assumed that only the robot’s actions change the cluttered scene. The examples for inference of the object logic states based on action feedback are shown in Table 1.

Combining the above-mentioned logic state of objects and the basic actions of robots, the fundamental steps of object segmentation in a cluttered scene are shown in Figure 10. This method is demonstrated by the example shown in Figure 11.

According to the result of the bounding box decomposition (Figure 11a), box 1 and box 2 may be an integrated object or two separate objects. Thus, the reasonable action is picking up the top box.

After executing the PickUp() action (Figure 11b), it can be determined whether the object was blocked by observing the variation of the point cloud according to Table 2, where the first rule can be used to find an overlaid object. After object 1 was grasped, if it was an integrated object, the vision system should no longer detect the point cloud in the object’s original position. If the point cloud is detected in that location, it indicates the point cloud is a new object 2, and that object 2 was overlaid with object 1 before the robot hand acted. The current pose of object 1 is determined based on the pose of the robot hand, and no visual algorithm is needed to re-detect it. The point cloud model of object 1 is updated according to the difference between the point cloud of object 1 at the previous moment and the point cloud of object 2 at the current moment, with a result shown in Figure 11c.

## 3. Grasp Planning

In general, there are many pose candidates for grasping a target object. Additionally, contact points between the object and the robot need to check if it is a stable grasp, such as with the force closure property. In this section, we describe an efficient grasp planning algorithm for high-quality and collision-free grasping pose generation. The complete grasp planning approach has two main aspects: The first is checking that the grasping pose has a collision-free path for a 6-DOF arm using rapidly exploring random trees (RRTs). The next is fast grasping quality analysis in wrench space.

### 3.1. Grasp Strategy

Like human beings grasp things, robot grasping movements can be defined as using one of two reach-to-grasp movements: top grasp or side grasp [12], as shown in Figure 12.

Humans may grasp contact points that are on the longer side of an object. Therefore, we decide the grasping direction according to the height of the bounding box of the target object. If the height is adequate, we can use a side grasp movement. Otherwise, we use a top grasp movement.

Thus, based on the OBB of the object, we can sample the position and orientation of the end-effector of the robot arm, as shown in Figure 13. Then, inverse kinematics (IK) are solved for the robot arm and a check is done for whether any collisions may occur between the robot arm system and the environment. If a collision occurs, then the grasping pose is resampled in other directions until the pose can avoid the collision. Next, we find a better grasp pose by looking at two criteria. One is the grasp pose can find a short collision-free path or the pose results in a high-quality grasp.

### 3.2. Path Planning

Path planning involves generally searching the path in a configuration space (*C*-space). For the robot arm, *C* means the workspace of the robot arm and each q∈C is a joint angle. Previous research has focused on only one goal for the purposes of motion path planning. However, there are multiple possible goals in the real world. The multi-objective RRT-Connect path planner was designed specifically for a real robot made as a high degrees of freedom (DOFs) kinematic chain [13].

The idea of multi-objective is illustrated in Figure 14. First, we set some goals in the configuration space where it is possible that some objectives are invalid, such as goal 3. Therefore, we need to check whether or not the goals are feasible according to the RRT-Connect algorithm. Let $T_{init}$ and $T_{goal}$ be the tree generated by $q_{init}$ and {$q_{goal}$}, respectively. The multi-objective RRT-Connect algorithm is given in Table 3.

The random Config() function generates random $q_{rand}\in C$. The extend function $T_{init}$ is illustrated in Figure 15. Each iteration attempts to extend the RRT by adding a new vertex that is biased by a randomly selected configuration. First, we find the vertex $q_{v}$ that is the nearest vertex to *q* in tree *T*. Then we extend the tree toward *q* from $q_{near}$ with ad-hoc variable distance *d* and assign a new variable $q_{newinit}$ before the collision event is checked.

Consider the robot arm joint space path planning. Figure 16 shows that with the same start configuration but different goal configurations, there will be a different path. The yellow object is the target object, and the white object is the obstacle which the robot arm and hand must avoid. The multi-objective RRT-Connect algorithm is used to search the six-dimensional *C*-space of the robot arm in order to find the collision-free path. Figure 16a shows the grasping pose that had a short collision-free path.

After searching for a path, we then needed to flatten, or prune, the path by removing any unnecessary waypoints, after which we can obtain a suitable trajectory for real robot grasping and manipulation. There are some efficient path pruning algorithms, like that shown in Ref. [14]. Even though the pruned path is not optimal, this method can remove most of the unnecessary waypoints very quickly, eliminating the chattering phenomenon in the RRT-Connect algorithm. Figure 17 shows the path before using the path pruning algorithm and the path after running the path pruning algorithm.

### 3.3. Grasp Analysis

After path planning is complete, the robot arm moves into position with a given orientation, and the robot hand performs continuous grasping until it touches the object. Here, the robot hand was treated as a simple gripper to reduce its number of DOFs, but a multi-fingered robot hand can adapt to the object better than a simple gripper because it has five fingers that can be independently adjusted to the object’s geometry. When all fingers are in contact with the object’s surface, it is still not clear whether or not the robot hand can firmly grasp the target object. Thus, grasp analysis is necessary. Grasp analysis is a matter of determining, given an object and a set of contacts, whether the grasp is stable using common closure properties.

A grasp is commonly defined as a set of contacts on the surface of an object [15,16,17]. The quality measure is an index that quantifies the effectiveness of a grasp. Force closure is a necessary property for grasping that requires a grasp to be capable of resisting any external wrench on the object, maintaining it in mechanical equilibrium [16].

One important grasp quality measure [18] often used in optimal grasp planning [19] assesses the force efficiency of a grasp by computing the minimum of the largest wrench that a grasp can resist, over all wrench directions, with limited contact forces [20], which equals the minimum distance from the origin of the wrench space to the boundary of a grasp wrench set.

The friction cone given by use of an equation is a convex cone, which consists of all non-negative combinations of the primitive contact force set $U_{i}$, defined as follows:(4)$$U_{i}=\left\{f_{i}|f_{n_{i}}=1,\;\sqrt{f_{t_{i}}^{2}+f_{o_{i}}^{2}}=\mu \right\},$$

The set Ui is the boundary of the friction cone. The image of Ui in the wrench space ℝ6 through the contact map Gi is called the primitive contact wrench set, which is expressed as:(5)Wi=Gi(Ui),

Let $W_{L_{1}}$ be the union of $W_{i}$ and $W_{L_{\infty }}$ be the union of the Minkowski sum for any choice of them:(6)$$W_{L_{1}}=\overset{m}{\underset{i=1}{\cup }}W_{i},\;W_{L_{\infty }}=\overset{m}{\underset{i=1}{⊕}}W_{i},$$

The grasp wrench set is defined to be the convex hull of $W_{L_{1}}$ or $W_{L_{\infty }}$, which is denoted by $W_{L_{1}}^{co}$ or $W_{L_{\infty }}^{co}$:(7)$$W_{L_{1}}^{co}=\left\{\sum_{i=1}^{m}G_{i}f_{i}|f_{i}\in friction\;cone,\;\sum_{i=1}^{m}f_{n_{i}}=1\right\},$$
(8)$$W_{L_{\infty }}^{co}=\left\{\sum_{i=1}^{m}G_{i}f_{i}|f_{i}\in friction\;cone,\;f_{n_{i}}\leq 1\right\},$$

Because the calculation of the Minkowski sum is more difficult, we chose the Equation (9) to compute the grasp wrench space (GWS). We can use some properties of GWS to evaluate the grasp quality. If the interior of WL1co is nonempty and contains the origin of the wrench space ℝ6, the grasp has force closure and is stable. Furthermore, the minimum of the largest resultant wrenches that a grasp can resist, over all wrench directions, with limited contact forces is an important quality measure of a force closure grasp. It can be formulated as the minimum distance between the origin of the wrench space ℝ6 and the boundary of WL1co, and the magnitude of the quality measure *Q* is defined as:(9)Q=minv∈WL1co‖v‖,
where *v* is ℝ6 vector. Figure 18 shows an example of the quality measure in the simulator. The negative score in Figure 18 means that the zero set {0} was not contained in the GWS. In other words, the grasp was unstable. For grasps with a positive score, the higher the score, the better the grasp’s capability of resisting a perturbation. Usually, *Q* ranges from 0.01 to 0.30 for stable grasping. We considered that grasps of more than 0.01 were sufficiently stable.

## 4. Real-Time Grip Force Selection and Control

The previous sections illustrated our method of estimating an object’s shape and determining a robust and safe grasping configuration. However, while executing the grasping motion, the appropriate force required to immobilize an object still needs to be set. The force regulation is also very important for a precision grasp, especially when grasping is followed by manipulation.

This paper presents a methodology for choosing an appropriate grasping force when a multi-finger robot hand grasps and lifts an object without knowing its weight, coefficient of static friction, or stiffness. To fulfill these requirements, a method that measures object stiffness and detects slippage by using a multi-sensor approach is proposed. The desired grasping force can be set with an upper limit to avoid damage and a lower limit to avoid slippage according to the object’s measured characteristics. Figure 19 shows the concept of a grip force selection structure.

The control flowchart shown in Figure 20 consists of two stages, the pre-grasp stage and the grasping stage. In the pre-grasp stage, the robot hand is positioned, and using the position controller, the robot hand fingers are closed on the target object until the normal force feedback is above a certain threshold. The threshold is chosen according to the stiffness measurement using a very small value in order to avoid damaging the object. Once all the fingers are in contact with the object, the position controller is stopped, and the hand enters the grasping stage. In the grasping stage, the robot hand will be switched to position-based force control.

### 4.1. Stiffness Measurement

When the robot hand is to grasp an object, the stiffness of the target object is a very important parameter that must be determined. Objects in our study were therefore divided into three categories according to their stiffness: a rigid object, a soft object, and a very soft object. Each category was defined by the upper limit grasp force.

A rigid object has high stiffness, and will not be deformed by the grasping force, such as a glass cup or metal bottle. For this category of objects, the upper limit force is defined as the maximum limit of the applied force of the robot finger.

Soft objects like plastic or paper cups have medium level stiffness and can be deformed when the robot hand increases the grasping force. For this category, the force that deforms objects is greater than the lift-off grasping force, so the appropriate upper limit force is set to have a minimal difference between the two forces.

The third category contains very soft objects such as sponges and other objects with little stiffness. This kind of object is greatly deformed by a slight grasping force. However, this deformation differs from soft objects because, in this case, it is acceptable for the robot hand to keep a stable grasp even if it causes a slight deformation of the object surface.

The concept of our method is that objects with different stiffness will have different mechanical responses to the initial contact. Thus, the force data to determine the stiffness is collected and processed. More details about our method will be discussed in the next subsection.

The stiffness measurement is taken using the following steps. First, the robot closes its fingers until it detects initial contact with the object. Once the hand contacts with the surface of the object, it continues to close for a fixed distance along the normal direction of the finger surfaces. Then the force curve is analyzed and features are selected to determine the object’s stiffness. Both three-axis force/torque (F/T) sensor [21] and tactile sensor arrays [5] are employed to collect the force data, and the obtained fusion results are shown in Figure 21. As seen from the force curve, different stiffness will have different force curves. The harder the object, the faster the force increases and the greater the steady force value when the loading is finished. What’s more, there are transient and steady states encountered during the stiffness measurement process. For the transient state, the hand makes contact with the object at a constant velocity. The results indicate that the force is proportional to contact speed for harder objects. Thus, the force curve slope $U_{d}$ was extracted as a feature to estimate the stiffness of a grasped object as:(10)$$U_{d}=\frac{1}{N_{1}}\sum_{k=i}^{k=N_{1}+i}F_{k},$$
where *i* is the time at which the hand starts to close, and $N_{1}$ is the sampling number of the transient state. $F_{k}$ is the normal force in the contact point at time *k*. For steady state, the object has been grasped. Further, the gripping force and the reverse force due to deformation have been balanced. According to Hooke’s law, *F* = *KX*, the steady pressure value is proportional to the stiffness of the object. Thus, the final steady pressure value of the curve can be used as a feature to estimate the stiffness of a grasped object:(11)$$U_{s}=\frac{1}{N_{2}}\sum_{k=j}^{k=N_{2}+j}F_{k},$$
where *j* is the time at which the hand stops, and $N_{2}$ is the sampling number of the steady state. Based on the above analysis, the characteristics of force during contact and steady-state force characteristics at equilibrium can be obtained.

After selecting the appropriate features, the k-nearest neighbors (K-NN) algorithm [22,23], was used to fuse these features and obtain more accurate stiffness recognition results. The K-NN algorithm is used to classify sample points into several distinct classes, and is summarized below:A positive integer *k* is specified along with a new sample.The *k* entries in the database that are closest to the new sample are selected.The most common classification of these entries is determined.This is the classification given to the new sample.

### 4.2. Slipping Detection and Avoidance

Robust slip detection ability [24] is one of the most important features needed for a grasping task. Knowledge about slip can help the robot prevent the object from falling down its hand. Furthermore, the sensation of slip is critical for a robot to grasp an object with minimal force [25].

In this study, we combined tactile sensor arrays [5] and three-axis force/torque (F/T) sensor [21] to obtain the slip signals. The framework of slip signal detection is shown in Figure 22. The slip feature based on frequency analysis and the slip feature based on motion estimation will be fused by support vector machine (SVM) algorithm.

A number of studies have developed a method that observes the frequency content of the contact forces when slippage occurs [26,27]. When the relative speed of the fingers and objects are relatively low, the theory of friction and vibration shows that the slippage is an intermittent vibration that causes vibration of the gripping force. Slippage can, therefore, be detected by the high-frequency signal it generates. Thus, a wavelet transform [28,29], was used to analyze and process the finger three-axis force/torque (F/T) sensor information and extract the real-time slip feature. Considering the complexity and real-time performance of the algorithm, the simplest Haar wavelet in wavelet transforms was adapted to analyze the sensor information. The Haar scaling function is defined as:(12)$$\phi (x)=\{\begin{array}{cc} 1, & if\;0\leq x<1\\ 0, & elsewhere \end{array},$$

The Haar wavelet function looks like:(13)ψ(x)=ϕ(2x)−ϕ(2x−1),

The sensor force data *f* (*t*)can be represented as the linear sum of the Haar scaling function and the Haar wavelet using Haar decomposition as:(14)$$f_{j}(t)=w_{j-1}+f_{j-1}(t),$$
where:(15)$$w_{j-1}=\sum_{k\in Z}b_{k}^{j-1}\psi (2^{j-1}t-k),$$
(16)$$f_{j-1}=\sum_{k\in Z}a_{k}^{j-1}\phi (2^{j-1}t-k),$$
(17)$$b_{k}^{j-1}=\frac{a_{2k}^{j}-a_{2k+1}^{j}}{2},\;a_{k}^{j-1}=\frac{a_{2k}^{j}+a_{2k+1}^{j}}{2},$$
and $a_{k}^{j-1}$ are called the detail coefficient. The Haar wavelet is employed to transform the contact force information into the slip process and to extract the slip signal by using the detail coefficient. It is known from the definition of the detail coefficient that it can be expressed by the difference of the force signals of the adjacent two moments.

Vibration-based methods can suffer from increased sensitivity to robot motion [30]. A very simple alternative is the tracking of the center of mass (the intensity weighted centroid) [31]. But this method is quite sensitive to noise and saturation effects. In addition, a shifting of weight is interpreted as a translation even if the pressure profile is not moving at all. Thus, we considered using methods based on the convolution calculation [32].

The overall workflow is shown in Figure 23. Let the pixel intensity function of a sensor matrix Tn at time step n be denoted as tn(x,y), Tn∈ℝM×N. The two-dimensional (2D) convolution of the discrete matrices, also known as complex-conjugate multiplication, is defined as:(18)tn−1(x,y)∗tn(x,y)=∑i=−∞∞∑j=−∞∞tn−1(y,j)tn(x−j,y−j),

This results in the convolution matrix *C* of size (2*M* − 1) × (2*N* − 1). The elements of *C* are therefore a measure of similarity between the two images, and the translation of the peak from the origin indicates the shift between them. In the context of slip detection, this relationship can be interpreted as a slip vector between two similar tactile sensor arrays profiles.

The decision is made to implement a slip detection algorithm by Alcazar and Barajas [33]. First, two index matrices A and B of the same size as the convolution matrix and consisting of repeating rows and columns, respectively, are defined as:(19)$$A=\left[\begin{array}{cccc} 1 & 2 & \cdots & 2N-1\\ 1 & 2 & \cdots & 2N-1\\ \vdots & \vdots & \ddots & \vdots \\ 1 & 2 & \cdots & 2N-1 \end{array}\right]-NB=\left[\begin{array}{cccc} 1 & 1 & \cdots & 1\\ 2 & 2 & \cdots & 2\\ \vdots & \vdots & \ddots & \vdots \\ 2M-1 & 2M-1 & \cdots & 2M-1 \end{array}\right]-M,$$

In the slip detection loop, the convolution matrix $C_{n-1}$ of the first pair of tactile matrices $T_{n-2}$ and $T_{n-1}$ is computed. A slip index along the *X* direction at time *T* is defined by:(20)$$\Delta x_{n-1}=E\left(\frac{A\cdot \mu_{c}^{T}}{sum(\mu_{c})}\right),$$

Similarly, a slip index along the *Y* direction at time *T* is computed by:(21)$$\Delta y_{n-1}=E\left(\frac{\mu_{r}^{T}\cdot B}{sum(\mu_{r})}\right),$$
where $\mu_{c}$ is a row vector containing the mean value of each column from the convolution matrix $C_{n-1}$. In contrast, $\mu_{r}$ is a column vector containing the mean value of each row from the convolution matrix $C_{n-1}$. *E*() and *sum*() denote the mean value and the sum of the elements of the vector, respectively. At the next time, the previous step was repeated. Again, the column and row means of defining the resulting convolution matrix $C_{n}$ and the displacements were computed. The final slip signal was computed with:(22)$$X_{n}=\sqrt{{\left(\Delta x_{n}-\Delta x_{n-1}\right)}^{2}+{\left(\Delta y_{n}-\Delta y_{n-1}\right)}^{2}},$$

In this study, we fused the two slip signals mentioned above along with the SVM to improve the identification accuracy of slip detection performance. The SVM is a machine learning method based on statistical learning theory (SLT) [34] that minimizes empirical risk and can, therefore, solve linear and nonlinear classification problems.

### 4.3. Real-Time Grasping Control

The purpose of the grasping control is to adjust the grasping force to prevent slippage and limit deformation of unknown objects. The controller must have a slippage adaptive function to adjust the grasping force in real time.

Let $F_{des}$ be the desired grip force (normal force) in each fingertip in a no-slip situation. Let *F_grip_* be the updated desired grip force that accounts for any slip conditions. Define *F_grip_* to be:(23)$$F_{grip}[k+1]=F_{grip}[k]+\alpha \beta ,$$
where α is a gain factor that is derived according to the object’s stiffness. A high stiffness object will have a bigger α, and this can help the hand quickly eliminate slippage. A low stiffness object will have a smaller α in order to avoid distortion of the object by increasing the grip force too quickly. In this paper, we have defined three levels of rigid categories, so there will be three corresponding α values. β is the slip signal derived from SVM results.

When slippage is detected, the regulated force must be applied almost instantaneously to ensure that the robot hand acts immediately. If the force is applied too slowly, the object will continue to slip and may even fall. Thus, a proportional-derivative (PD) position controller was used to achieve stable grasping control as follows:(24)$$U=\left\{\begin{array}{cc} K_{p}(F_{grip}-F)+K_{d}(\frac{dF}{dt}), & if\;F\leq F_{\mathrm{limit}}\\ 0, & if\;F>F_{\mathrm{limit}} \end{array}\right\},$$
where *U* is the output of position controller that determines each fingertip’s normal direction in Cartesian space, *K_p_* and *K_d_* are the proportional and derivative parameters, and *F* is the real feedback force from the sensors in the fingertips. *F*_limit_ is set according to the object’s stiffness.

## 5. Experiment Results

This section describes how the NTU 6-DOF robot arm [5,6,35,36,37] and the NTU five-finger robot hand were equipped with additional hardware and software to enable the resultant grasp of unknown objects. We combined the robot arm and robot hand to facilitate planning and real-time control, as shown in Figure 24.

This paper presents the on-going research of the National Taiwan University (NTU) five-finger robot hand. The purpose of the development of the NTU five-finger robot hand is for delicate dynamic grasp. To design a motor-driven, dexterous robot hand, we analyzed the human hand. Our design features a customized fingertip three-axis force/torque (F/T) sensor and joint torque sensors integrated into each finger, along with powerful super-flat brushless DC (BLDC) motors (The MAXON (Sachseln, Switzerland) motor measures only 23 mm in the outer diameter and creates a unit that is only 18.5 mm in height, with a motor weight of just 15 g. The rated speed and torque of the motor idle at 4140 rpm; the 3*W* motors are available in a 9*V* version and provide a maximum torque of 8.04 mNm. The motors include digital Hall sensors and digital magnetic Encoders. A linear per position resolution of 0.001 mm is obtained by coupling a MAXON Encoder MR (Type M 512 cpt) to the motor. and tiny harmonic drivers (HD) (Harmonic Drive™ gear AG, Limburg an der Lahn, Germany, HDUC-5-100-BLR, gear ratio 1:100). By using a steel coupling mechanism, the phalanx distal’s transmission ratio is exactly 1:1 in the whole movement range. The rest of this paper presents a control strategy for the NTU five-finger robot hand. For robust grasp, we implemented classical impedance control. Our goal is to develop 21 DOFs dexterous robot hand. The hand has an independent palm and five identical modular fingers; each finger has three DOFs and four joints. We designed this robot hand by improving previously developed robot hand designs and by analyzing actual human hand motions. This NTU motor-driven, the five-finger robot hand can be equipped with a three-axis force/torque sensor [21] at each fingertip, as well as with developed distributed tactile sensor arrays with 376 detecting points on its surface [5]. The hand can communicate with external sources using technologies such as EtherCAT and controller area networks (CAN bus). To evaluate the performance of this robot hand design, we analyzed workspace, intersection volume, and manipulability, as shown in Figure 24.

A dexterous robot hand needs at a minimum a set of force and position sensors to enable control schemes like position control and impedance control in autonomous operation and teleoperation. The aim of the sensory design is to integrate into the artificial hand a great number of different sensors in order to confer to the hand functionalities similar to that of a human hand. Sensor equipment for the NTU five-finger robot hand is shown in Table 4.

The maximum payload of the NTU 6-DOF robot arm is over two kg. The specification of the NTU 6-DOF robot arm is shown in Table 5.

Because there are two kinds of the sensor on the fingertip, a three-axis F/T sensor and tactile sensor arrays, we calibrate and fuse the two sensors’ normal force. We use weight groups (20, 50, 70, 100, 120, 150, 170 and 200 g) as the standard value and record the error between the measured value of the two sensors’ normal force and weight’s standard value. The result is shown in Figure 25. From the figure, in the range of 0 to 100 g, the linearity of the two sensors is different, while in the range of 100 to 200 g, the measured values of the two sensors are quite similar.

Therefore, we fuse the sensor data in the two areas separately using the least squares method. The result is shown in Figure 26. Let $f_{1}$ and $f_{2}$ denote three-axis F/T sensor and tactile sensor arrays measurements of normal force, respectively. In the 0–0.98 N range, the fusion force is $F=1.8413f_{1}+3.6176f_{2}-0.0245$. In the 0.98–1.96 N range, the fusion force is $F=4.9991f_{1}-1.2253f_{2}+0.1103$. The results show that the fused sensor values have good linearity with the standard value in the whole range.

### 5.1. Experiment 1: Slipping Detection and Avoidance

This experiment was mainly aimed at verifying the accuracy of slip detection and the quick response of dynamic adjustment to the grasping force.

In order to collect the sensor data during the slip, we set up the experiment, as shown in Figure 27. First, we give the fixed initial grasp pose and grip force. When an initial steady grip was reached, a heavy load was added to the object (rice in the cup), as shown in Figure 28a. As the weight increased, the object slipped (Figure 28b), so the additional weight was used as a disturbance that caused relative slippage. Finally, the object slipped from the hand (Figure 28c), and we recorded the three-axis force/torque (F/T) sensor and tactile sensor arrays data.

The tangential force collected by the three-axis force/torque (F/T) sensor in the thumb is shown in the upper graph of Figure 29. From this figure, we can see that the raw data contained high-frequency random noise. In order to remove the noise, we used a low-pass filter to process the raw data. The result after filtering is shown in the bottom graph of Figure 29. The upper graph in Figure 30 presents the discrete wavelet transform from the finger three-axis force/torque (F/T) sensor. The bottom graph shows the displacement value from the tactile sensor arrays data. When t = 1.28 s, slippage occurred.

The slip experiment dataset is shown in Figure 31, where the red spots represent the slip situation, and the blue spots represent the no-slip situation. Then we used LIBSVM [38] to train the model. In this study, we used a grid search method to find the best parameter (c, g) for an RBF kernel. As shown in Figure 32, we found that the best parameter was (8, 32), with a cross-validation rate of 97.5%.

Next, we used the SVM predict function to help the robot hand detect the slip signal in real time. Similar to the previous experiment, the task of this experiment was to grasp the plastic cup with three fingers and gradually add rice to the cup. This time, we used slip detection to adjust the grip force in time to suppress the slip. The experimental process is shown in Figure 33. In this experiment, when an initial steady grip was reached, a heavy load was added to the object (rice in the cup) (Figure 33a–c). The additional weight was used as a disturbance that caused relative slippage, we succeeded in avoiding the slippage (Figure 33d–f). However, because we had not measured the stiffness of the object, when the robot hand came into contact with the object (heavy load), it caused deformation of the plastic cup (Figure 33g–i). The slip signal and the grip force change are shown in Figure 34 and Figure 35.

### 5.2. Experiment 2: Grasping in a Cluttered Scene

This experiment demonstrated grasping in a cluttered scene. The experimental environment used is shown in Figure 36. The objects were set on the blue table, and the robot hand-arm system was located at one end of the table. Two depth sensors were placed on one end of the table and on the side of the robot hand-arm system to capture the point cloud of the environment.

Because the NTU robot hand is a right hand, we predefined our task as being to clear the table from the right side to the left side. The first step was to grasp the milk tea. In the robot simulator block, the simulator chose the target object (milk tea). Then, the grasp strategy was used to find the grasp configuration of the robot hand-arm. The path planning used the RRT-Connect algorithm to find collision-free paths for grasping, as shown in Figure 37b.

Once the hand contacted the object, it first took a rigid measurement, as shown in Figure 38a. According to the result of K-NN (Figure 38b), the first object belonged to the low stiffness category. The planner generated a collision-free path from the current position to the target box area (Figure 37c), and then, in the grasping stage, the robot hand turned to position-based force control. According to the multi-sensor feedback as well as the slip detection and stiffness measurement, the robot hand could grasp the object successfully and place it in a specific area according to the object’s stiffness. The procedure used for the real robot hand-arm system is shown in Figure 39.

Next, we found that in the middle of the table, there were three objects stacked together, but the vision system considered them to be one object only. As shown in Figure 40a, the target object was approximated by using a few bounding boxes, and the robot hand is first considered grasping the top box. The grasp poses are shown in Figure 40b. Similar to grasping object 1, the hand determined the stiffness measurement (Figure 41a), and K-NN indicated that object 2 belonged to the medium stiffness category. The procedures for the simulation and the real robot hand-arm are shown in Figure 40 and Figure 42, respectively.

The vision system collected the point cloud in the scene again (Figure 43a) to update the object state according to the object state inference. After grasping object 2, there was still a point cloud in the same location, meaning there were multiple objects stacked in the same place, and the robot hand considered the next object remaining on the table to be object 3. Grasp planning was performed in a similar manner as the previous objects. The stiffness measurement (Figure 44a) and K-NN result (Figure 44b) indicated that object 3 belonged to the category of high stiffness. The procedures for the simulation and the real robot hand-arm are shown in Figure 43 and Figure 45, respectively. Finally, the last object in the original location was considered to be object 4 is shown in Figure 46. The stiffness measurement (Figure 47a) and K-NN result (Figure 47b) indicated that object 4 also belonged to the high stiffness category. As the robot interacted with them, all the real objects were gradually separated. The procedures for the real robot hand-arm system and the simulation are shown from Figure 46, Figure 47 and Figure 48.

The last object on the table, in a different location, was a cloth puppet, for which the bounding box is shown in Figure 49a. Based on this bounding box, the grasp planner chose an appropriate grasp configuration. According to the result of K-NN (Figure 50b), object 5 belonged to the low stiffness category. The procedures for the real robot hand-arm system and the simulation are shown in Figure 49 and Figure 51, respectively. The classification results are shown in Figure 52.

As also shown in Figure 52, all the objects on the table were successfully separated and placed in different boxes based on the stiffness of the object.

## 6. Conclusions

The aim of those experiments was to train a robot hand-arm system to grasp unknown objects. Grasping an object is a simple task for a human, who can grasp and manipulate any object whether he has seen it or not. We want a robot that can work in a human environment in a similar manner to grasp a variety of objects of different materials, shapes, and sizes in a cluttered scene. Therefore, we focused on two issues. The first issue was the methodology for setting the real-time grasping force of the fingers when grasping an object with unknown stiffness, weight, and friction. The second issue was on-line sorting objects in a cluttered scene.

When the task is to grasp an unknown object, it is the most concern to avoid dropping it due to slipping, which is considered a serious error because of breakage of the object or unpredictable disasters. What is more, crushing objects or grasping them with excessive force should also be avoided. In order to grasp an unknown object with an appropriate force, we used multi-sensor inputs and sensor fusion technology to detect slippage and measure the stiffness of objects. The experiments demonstrated that multi-sensor information can ensure that the robot hand grasp unknown objects safely and make the robot hand’s motion more like that of a human.

Physically interacting to improve the understanding of an environment is a natural behavior for humans, and based on this, we proposed an algorithm that combines perception and manipulation to enable a robot to sort objects in a cluttered space according to one specific property (in this case, stiffness). In contrast to pure visual perceptual approaches, our method can quickly rearrange objects. Our experiments show that sorting is more viable and reliable when using this method to deal with a quantity of unknown objects.

## Figures and Tables

**Figure 1 sensors-19-01595-f001:**
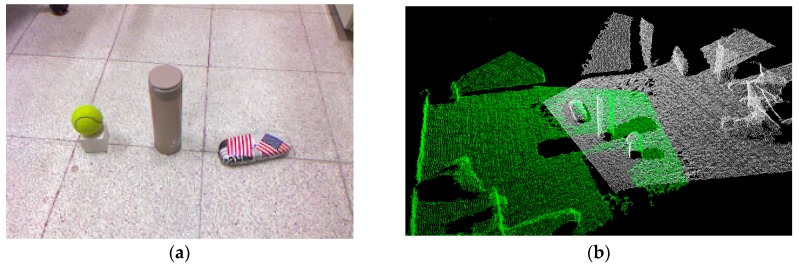
Real environment and point cloud: (**a**) Real environment; (**b**) Point cloud from the depth sensor.

**Figure 2 sensors-19-01595-f002:**
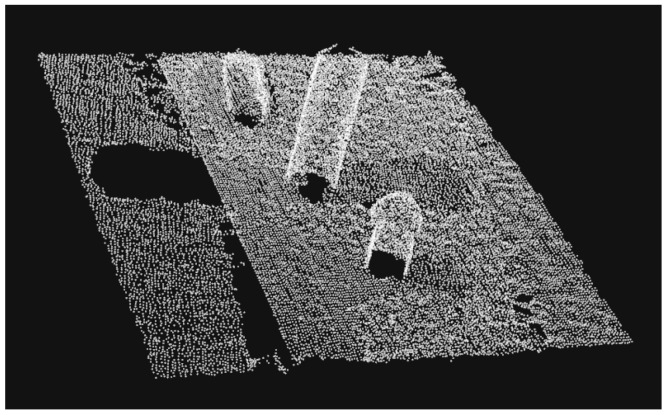
The output of pass-through filter.

**Figure 3 sensors-19-01595-f003:**
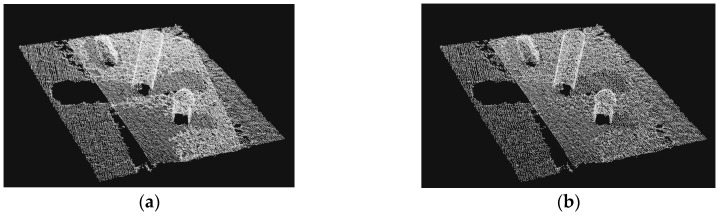
Down-sampling: (**a**) Original point cloud; (**b**) After down-sampling.

**Figure 4 sensors-19-01595-f004:**
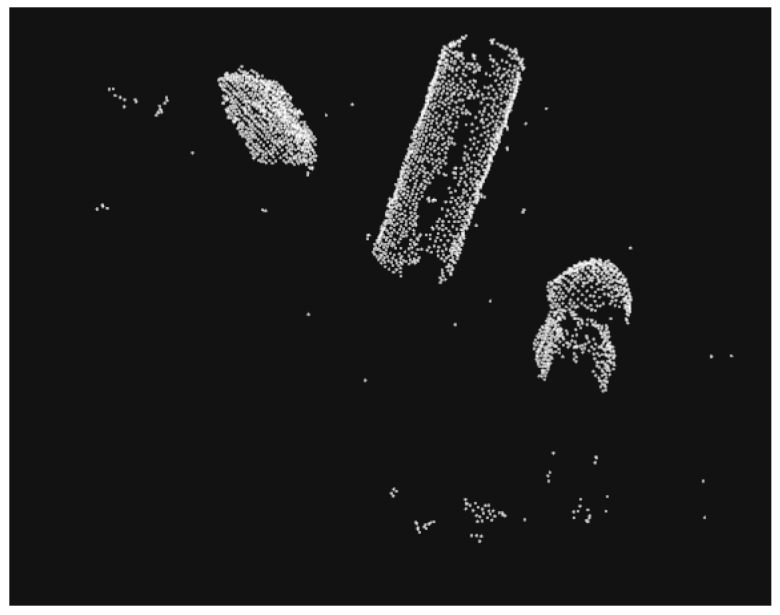
Objects after removing the ground.

**Figure 5 sensors-19-01595-f005:**
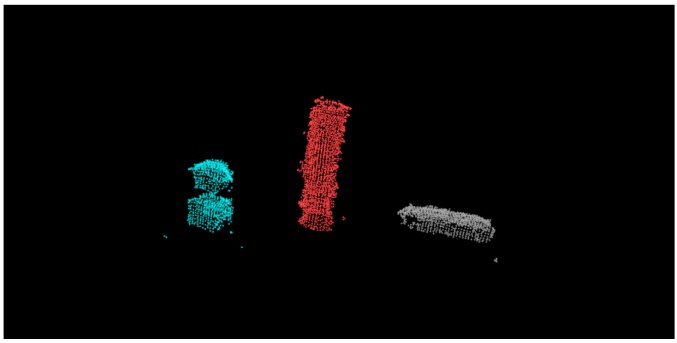
Point cloud after performing segmentation.

**Figure 6 sensors-19-01595-f006:**
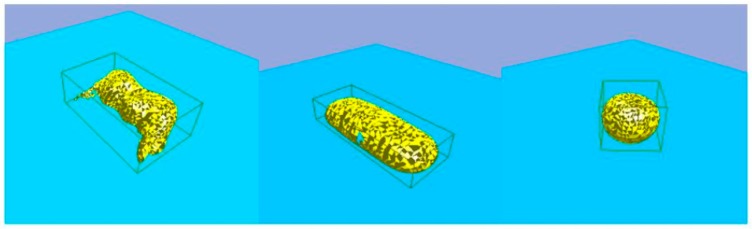
The bounding box of a set of objects.

**Figure 7 sensors-19-01595-f007:**
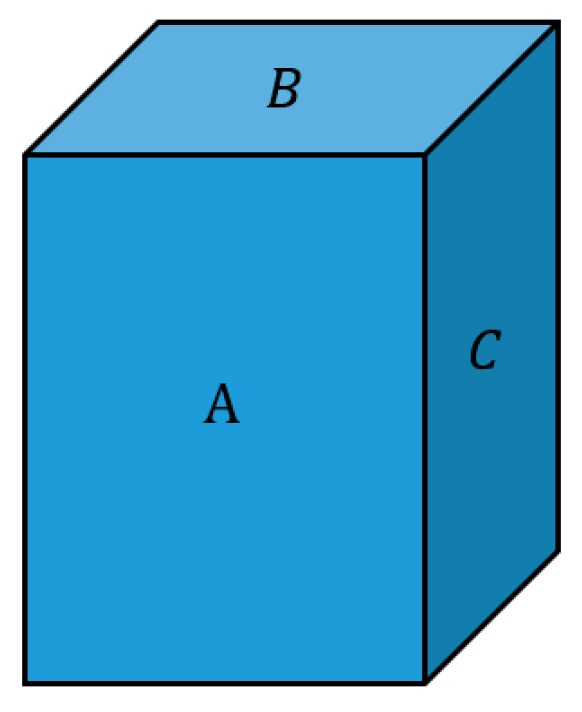
Parallel box cutting planes.

**Figure 8 sensors-19-01595-f008:**
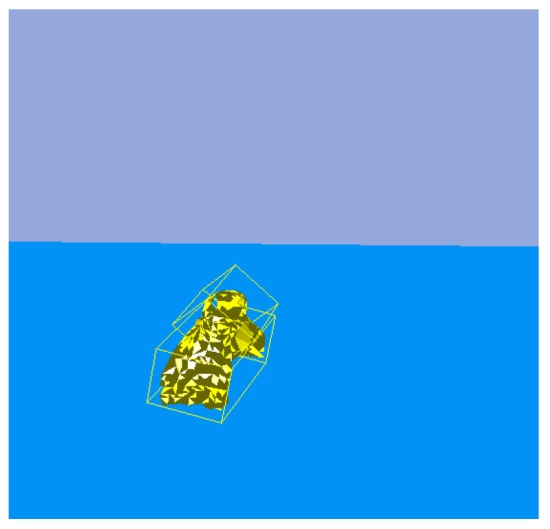
Best cuts.

**Figure 9 sensors-19-01595-f009:**
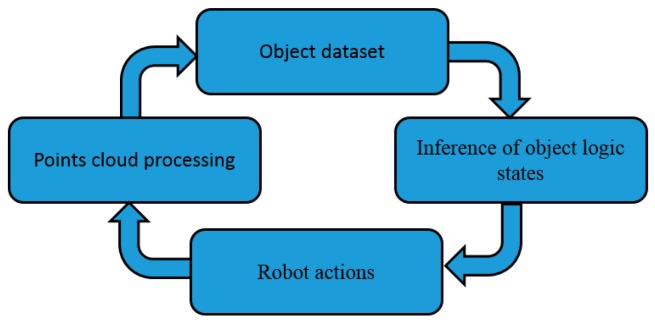
The framework of unknown object segmentation.

**Figure 10 sensors-19-01595-f010:**
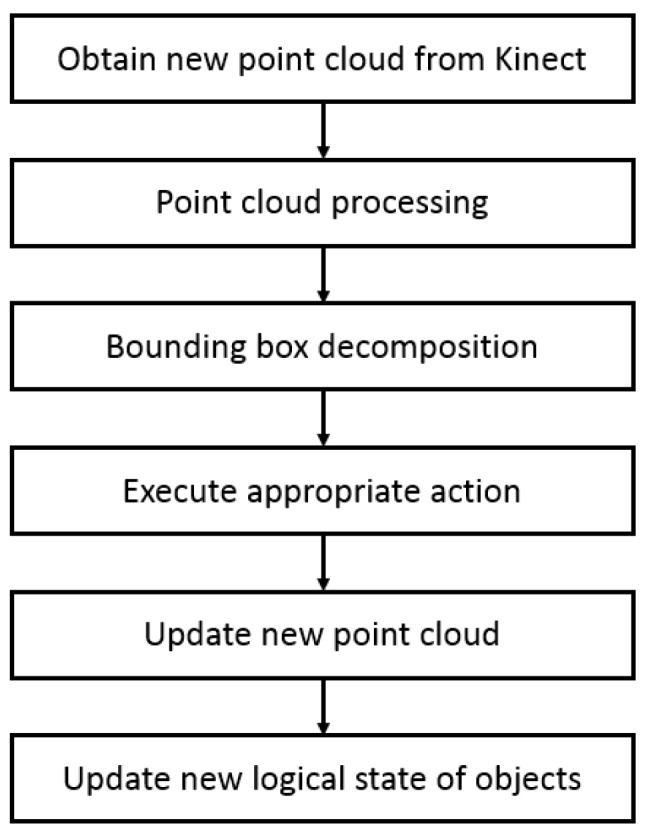
The flow of object segmentation.

**Figure 11 sensors-19-01595-f011:**
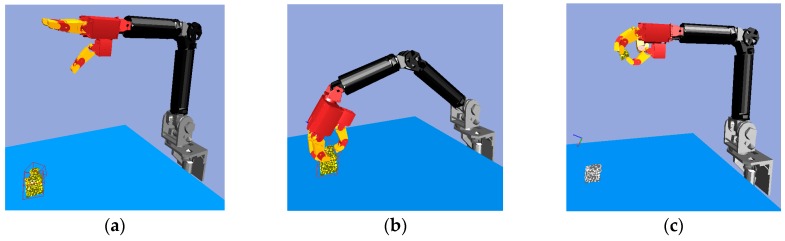
Object segmentation in the simulator: (**a**) Object’s original position; (**b**) Previous moment; (**c**) Current moment.

**Figure 12 sensors-19-01595-f012:**
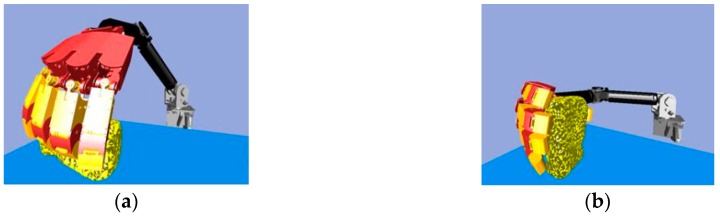
The robot grasping movements: (**a**) Top grasp; (**b**) Side grasp.

**Figure 13 sensors-19-01595-f013:**
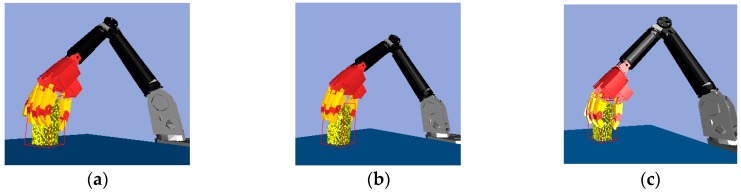
The sampling of the robot arm: (**a**) Top grasp movement; (**b**) A short collision-free path; (**c**) High-quality grasp.

**Figure 14 sensors-19-01595-f014:**
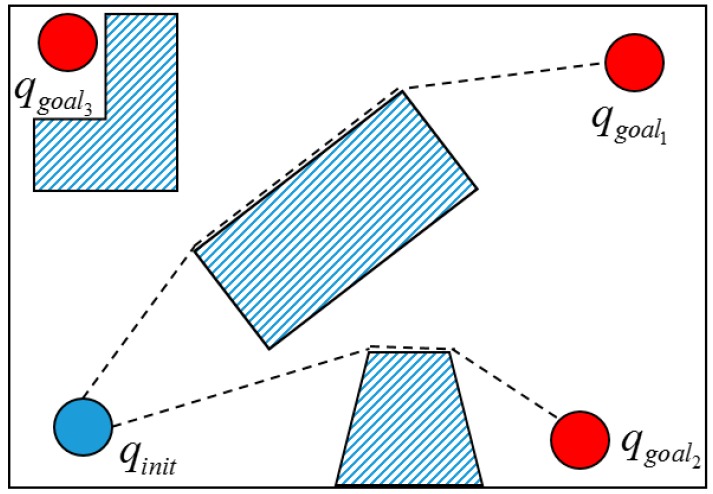
Multi-goal planner.

**Figure 15 sensors-19-01595-f015:**
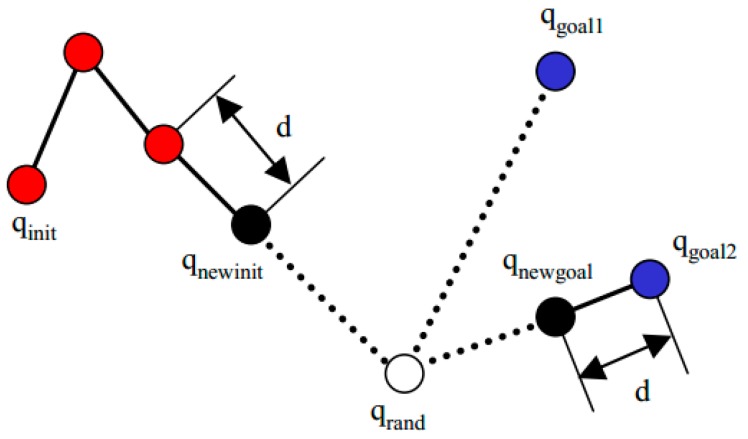
The extend function.

**Figure 16 sensors-19-01595-f016:**
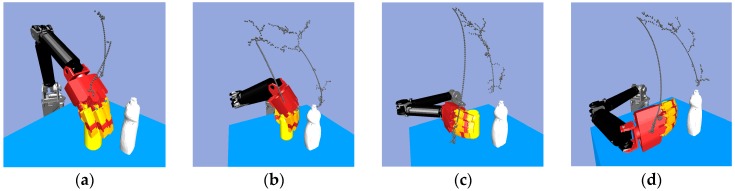
Multi-objective RRT-Connect in the simulator: (**a**) Path size: 59; (**b**) Path size: 238; (**c**) Path size: 191; (**d**) Path size: 231.

**Figure 17 sensors-19-01595-f017:**
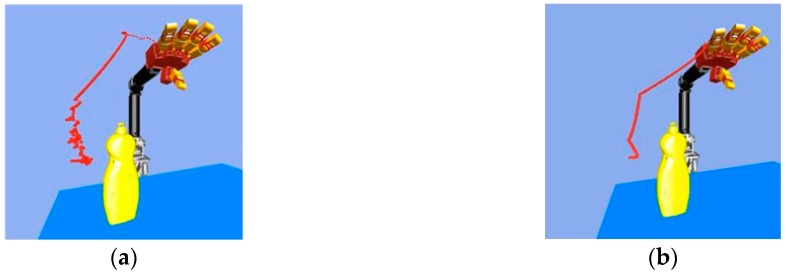
Path pruning algorithm: (**a**) Original path; (**b**) The path through path pruning algorithm.

**Figure 18 sensors-19-01595-f018:**
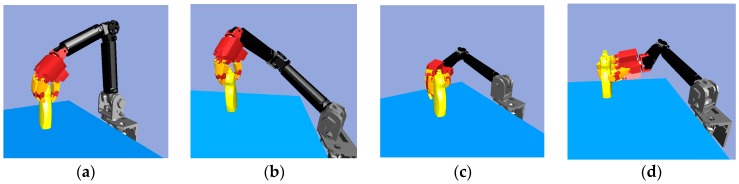
Quality measure example for a bottle: (**a**) Path size: 0.0135; (**b**) Path size: 0.001; (**c**) Path size: 0.017; (**d**) Path size: −0.27.

**Figure 19 sensors-19-01595-f019:**
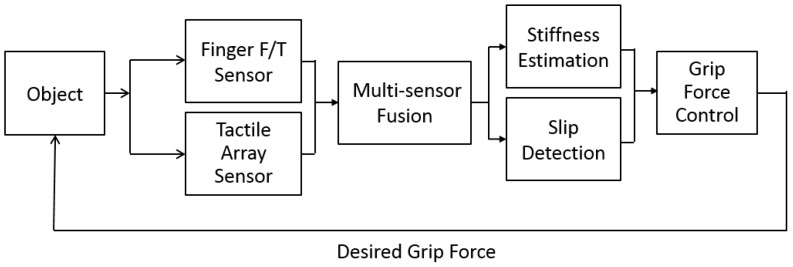
Grip force selection structure.

**Figure 20 sensors-19-01595-f020:**
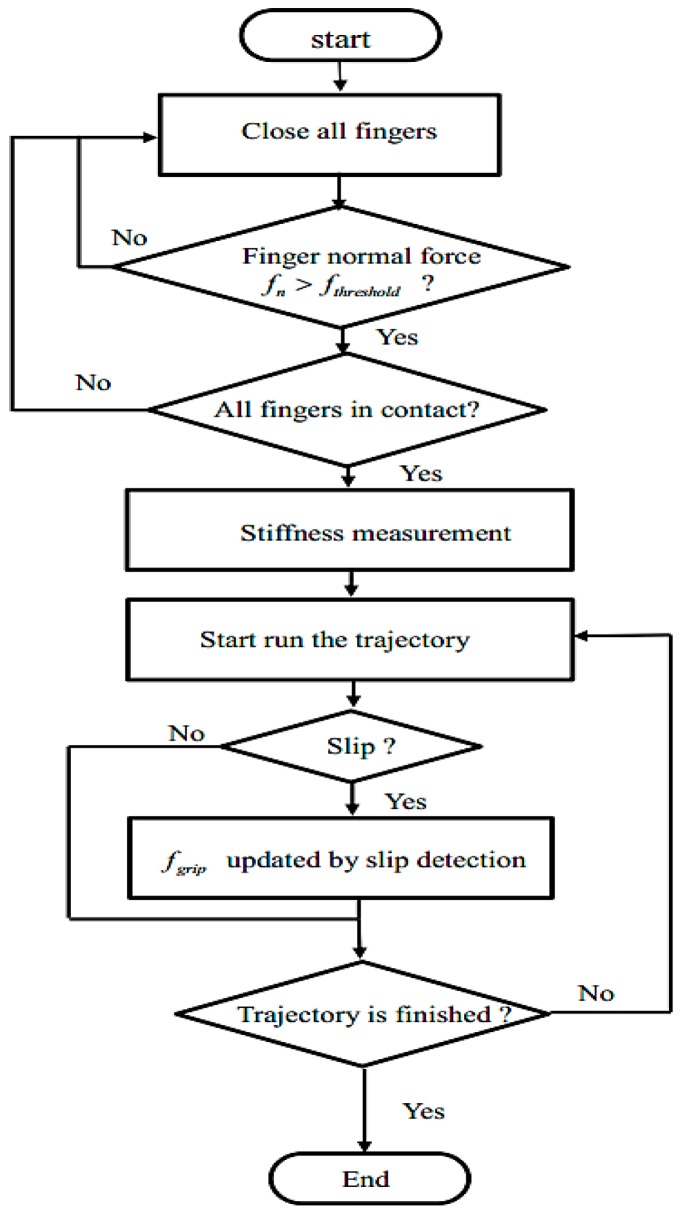
Control diagram of the grip controller.

**Figure 21 sensors-19-01595-f021:**
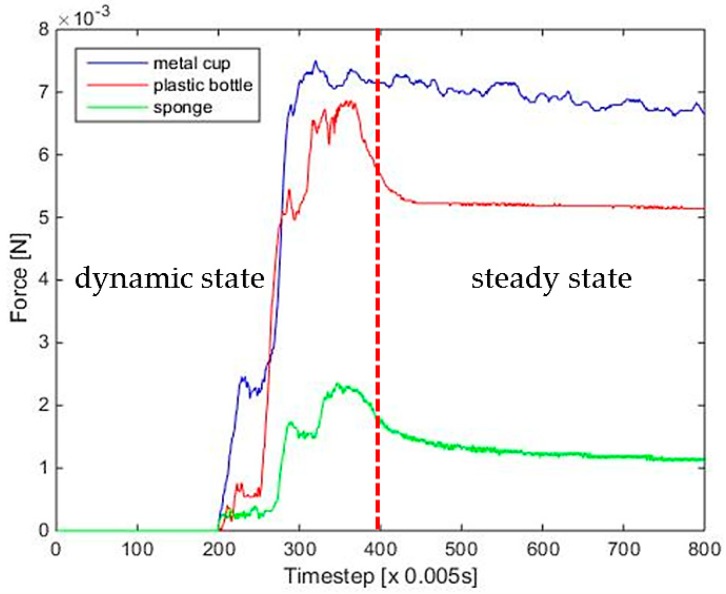
Force curves of different categories of objects.

**Figure 22 sensors-19-01595-f022:**
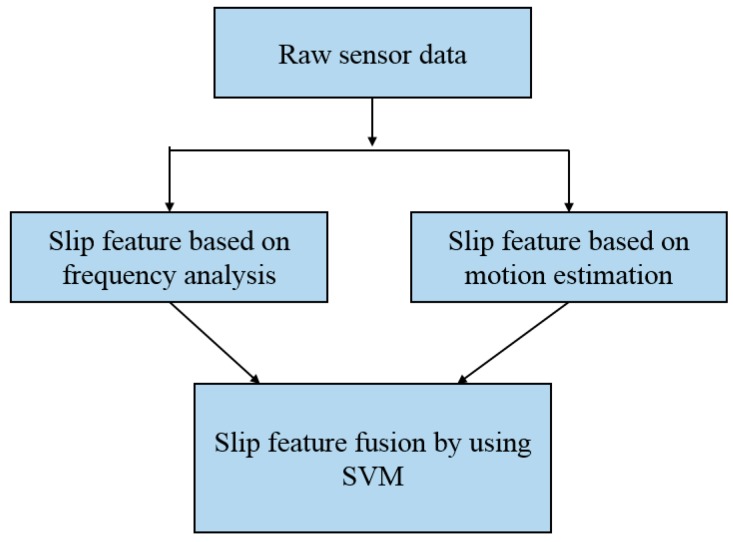
Slip detection and avoidance framework.

**Figure 23 sensors-19-01595-f023:**
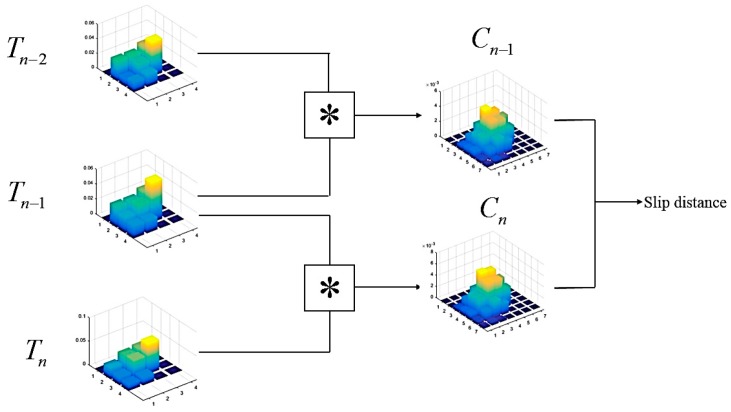
Slip detector based on motion estimation workflow.

**Figure 24 sensors-19-01595-f024:**
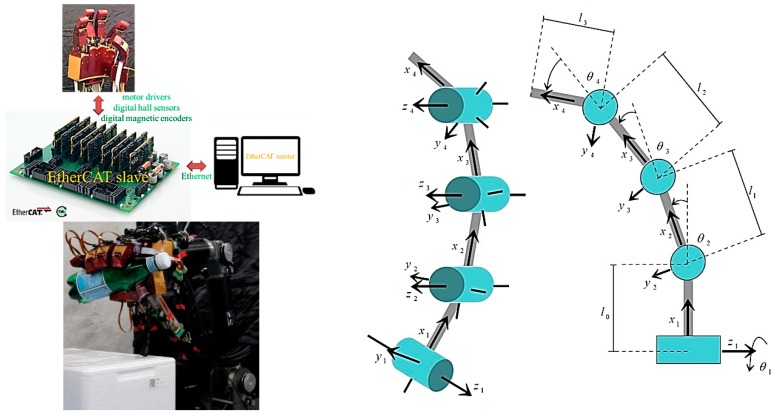
NTU five-finger hand with modular fingers (Four link manipulator to model one finger of the robot hand).

**Figure 25 sensors-19-01595-f025:**
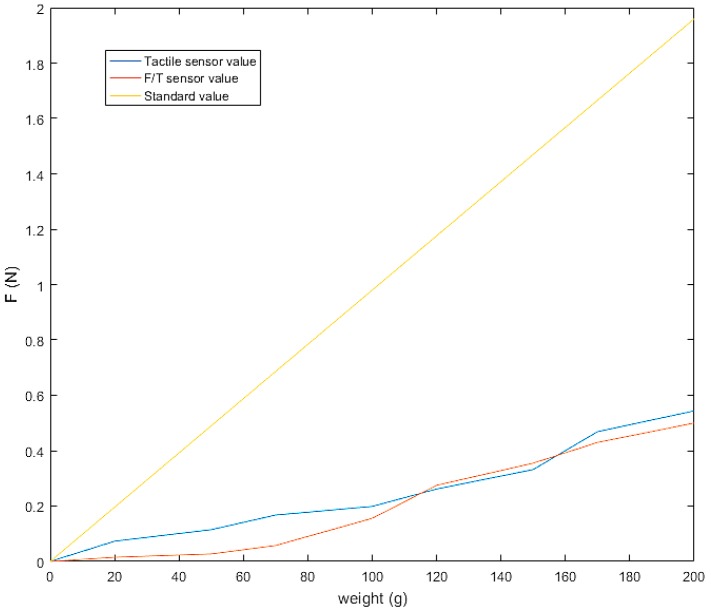
Force sensor calibration.

**Figure 26 sensors-19-01595-f026:**
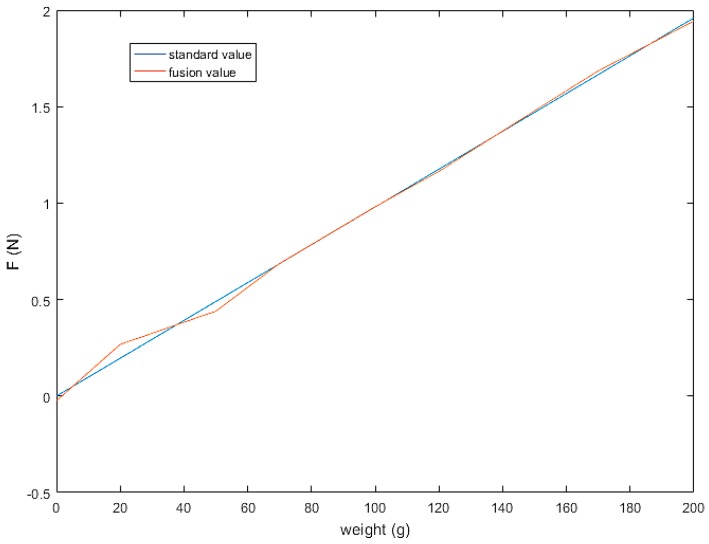
Sensor fusion result.

**Figure 27 sensors-19-01595-f027:**
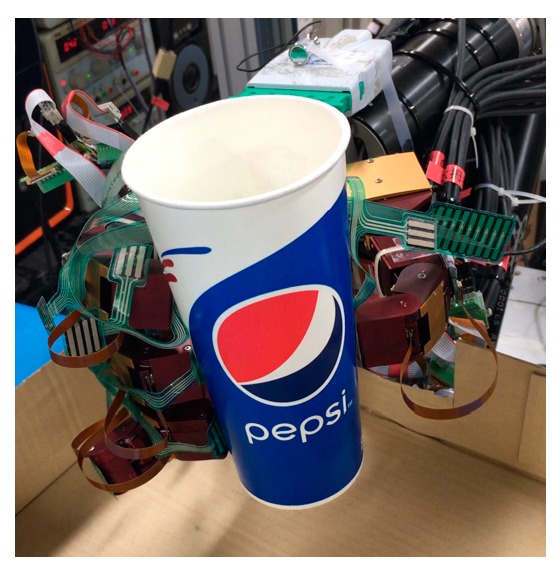
Experiment setup for slip detection.

**Figure 28 sensors-19-01595-f028:**
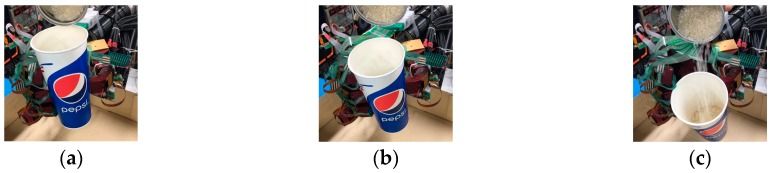
The process of slipping: (**a**) The heavy load was added to the object (rice in the cup); (**b**) Weight increased; (**c**) The object slipped from the hand.

**Figure 29 sensors-19-01595-f029:**
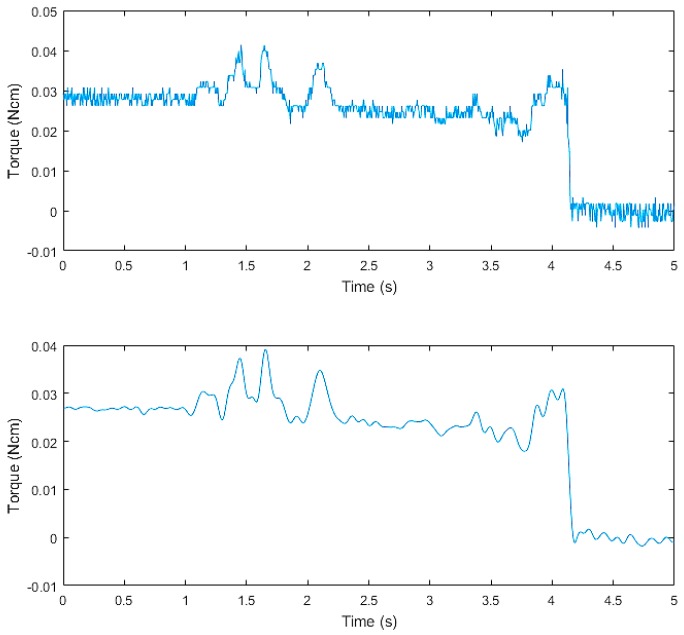
The tangential force of finger three-axis force/torque (F/T) sensor.

**Figure 30 sensors-19-01595-f030:**
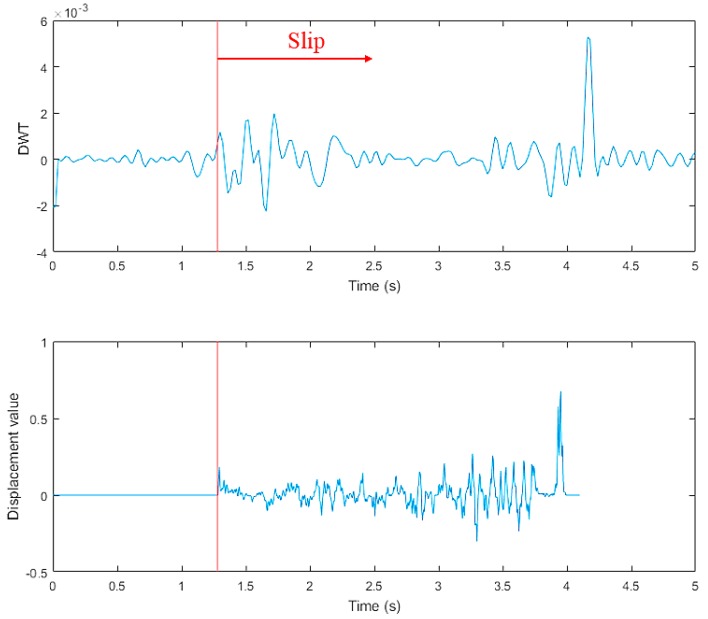
Results of slip detection experiment.

**Figure 31 sensors-19-01595-f031:**
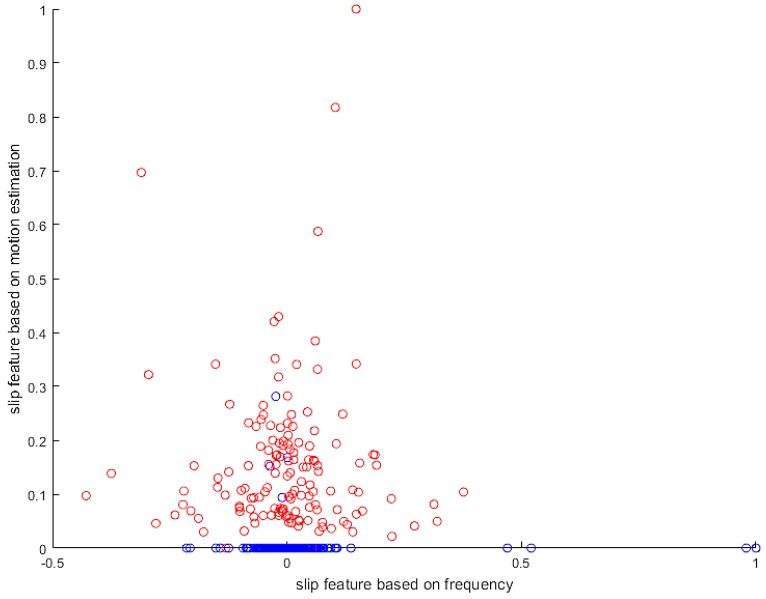
Dataset.

**Figure 32 sensors-19-01595-f032:**
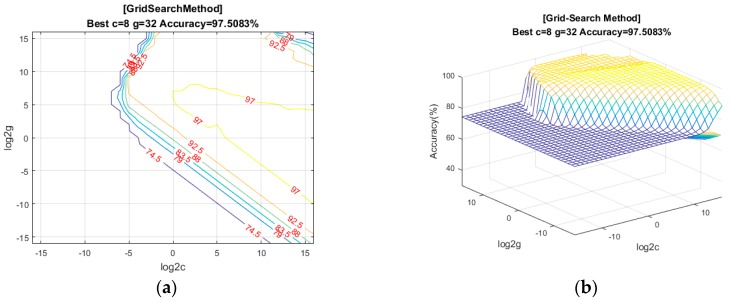
The result of the grid search method: (**a**) Grid-Search Method; (**b**) The cross-validation rate.

**Figure 33 sensors-19-01595-f033:**
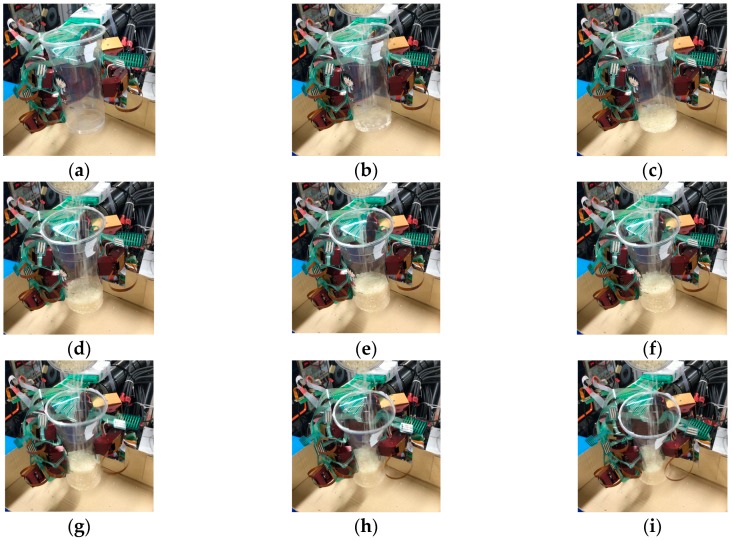
Slip prevention experiment: (**a**–**c**) When an initial steady grip was reached, a heavy load was added to the object (rice in the cup); (**d**–**f**) The additional weight was used as a disturbance that caused relative slippage, we succeeded in avoiding the slippage; (**g**–**i**) Finally, when the robot hand came into contact with the object (heavy load), it caused deformation of the plastic cup.

**Figure 34 sensors-19-01595-f034:**
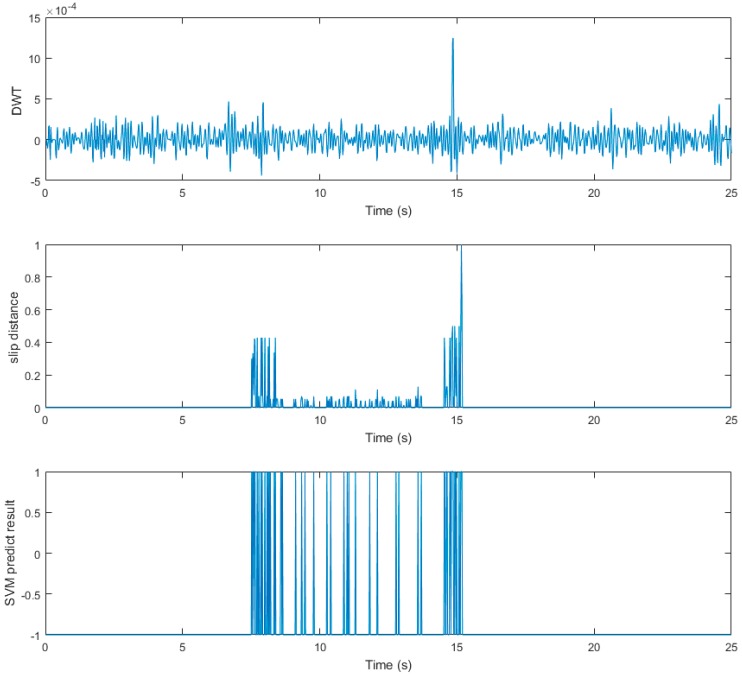
Slip signal.

**Figure 35 sensors-19-01595-f035:**
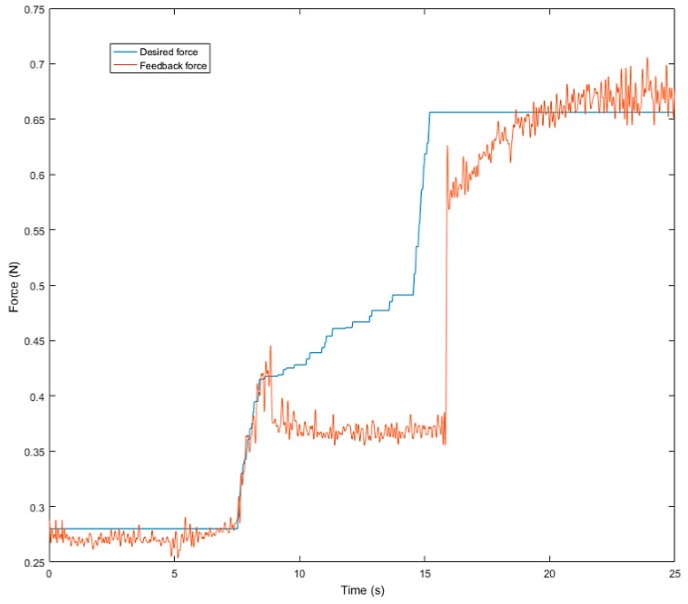
Grip force.

**Figure 36 sensors-19-01595-f036:**
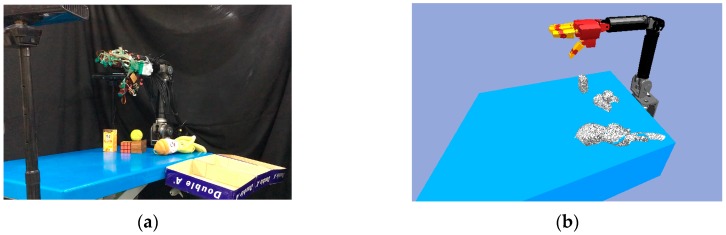
The experimental environment: (**a**) Real environment; (**b**) Objects in the simulator.

**Figure 37 sensors-19-01595-f037:**
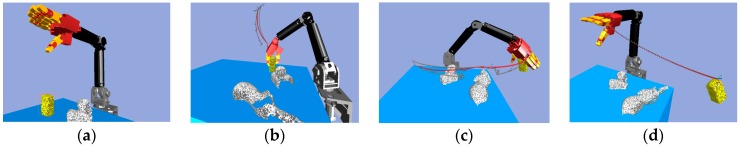
Experiment 2 in a simulator: (**a**) The target object (milk tea); (**b**) Collision-free paths for grasping; (**c**) RRT-Connect algorithm; (**d**) Successfully separated and placed in different boxes.

**Figure 38 sensors-19-01595-f038:**
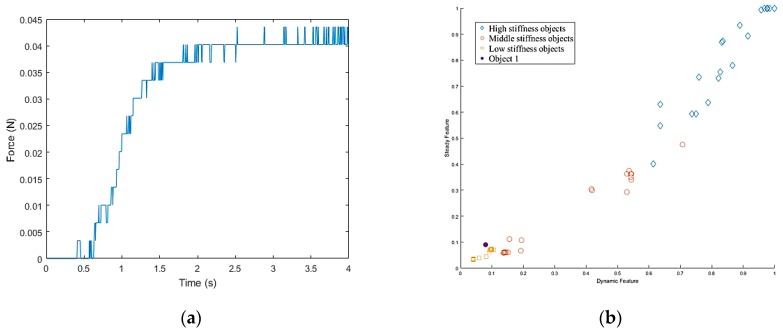
Object 1 stiffness measurement: (**a**) Stiffness; (**b**) K-NN.

**Figure 39 sensors-19-01595-f039:**
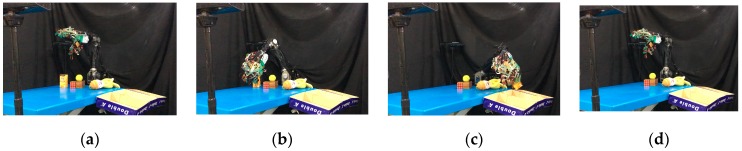
A procedure of grasping the first object: (**a**) The target object (milk tea); (**b**) Collision-free paths for grasping; (**c**) RRT-Connect algorithm; (**d**) Successfully separated and placed in different boxes.

**Figure 40 sensors-19-01595-f040:**
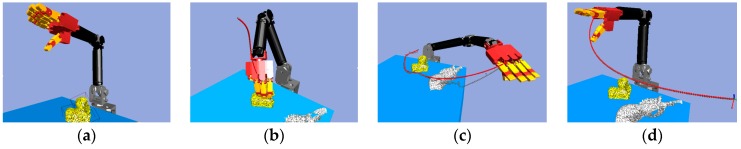
A procedure of grasping the second object in the simulator: (**a**) The target object (top box); (**b**) Collision-free paths for grasping; (**c**) RRT-Connect algorithm; (**d**) Successfully separated and placed in different boxes.

**Figure 41 sensors-19-01595-f041:**
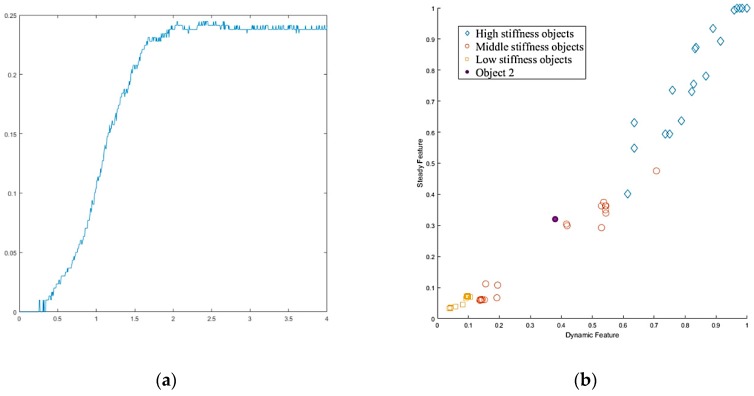
Object 2 stiffness measurement: (**a**) Stiffness; (**b**) K-NN.

**Figure 42 sensors-19-01595-f042:**
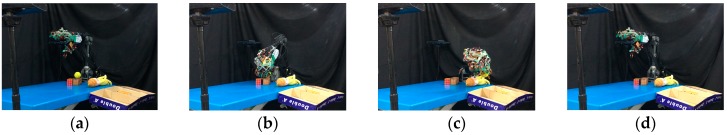
A procedure of grasping second object: (**a**) The target object (top box); (**b**) Collision-free paths for grasping; (**c**) RRT-Connect algorithm; (**d**) Successfully separated and placed in different boxes.

**Figure 43 sensors-19-01595-f043:**
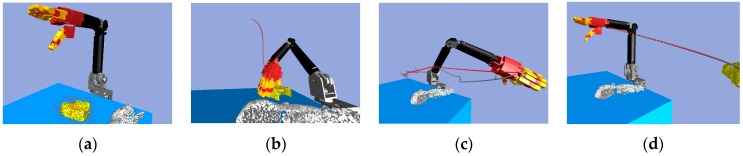
A procedure of grasping the third object in the simulator: (**a**) The target object; (**b**) Collision-free paths for grasping; (**c**) RRT-Connect algorithm; (**d**) Successfully separated and placed in different boxes.

**Figure 44 sensors-19-01595-f044:**
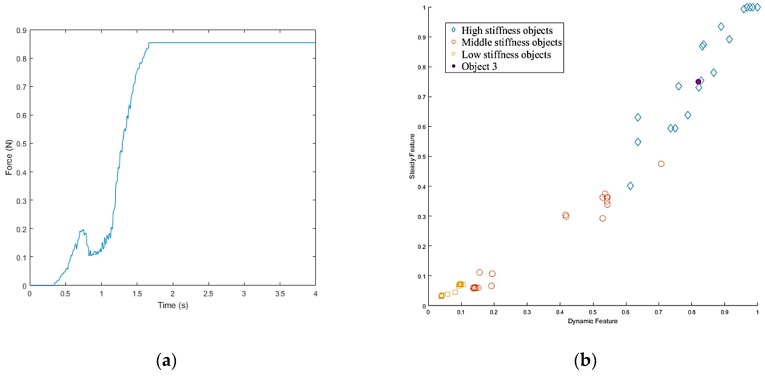
Object 3 stiffness measurement: (**a**) Stiffness; (**b**) K-NN.

**Figure 45 sensors-19-01595-f045:**
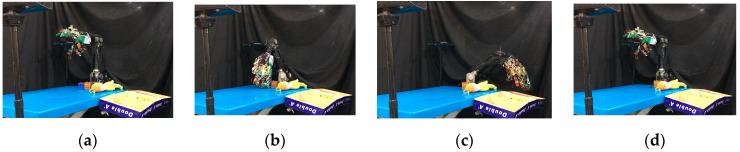
A procedure of grasping the third object: (**a**) The target object; (**b**) Collision-free paths for grasping; (**c**) RRT-Connect algorithm; (**d**) Successfully separated and placed in different boxes.

**Figure 46 sensors-19-01595-f046:**
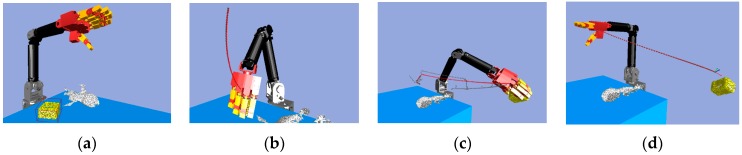
A procedure of grasping the fourth object in the simulator: (**a**) The target object; (**b**) Collision-free paths for grasping; (**c**) RRT-Connect algorithm; (**d**) Successfully separated and placed in different boxes.

**Figure 47 sensors-19-01595-f047:**
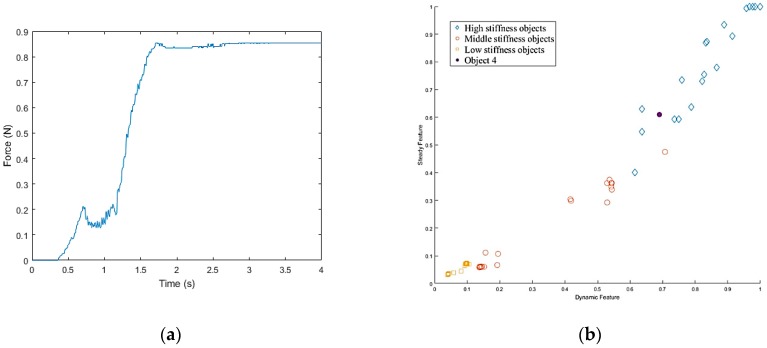
Object 4 stiffness measurement: (**a**) Stiffness; (**b**) K-NN.

**Figure 48 sensors-19-01595-f048:**
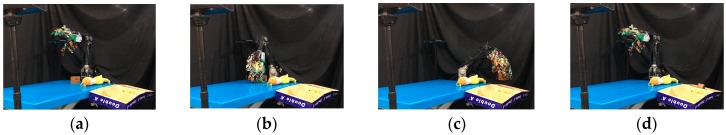
A procedure of grasping the fourth object: (**a**) The target object; (**b**) Collision-free paths for grasping; (**c**) RRT-Connect algorithm; (**d**) Successfully separated and placed in different boxes.

**Figure 49 sensors-19-01595-f049:**
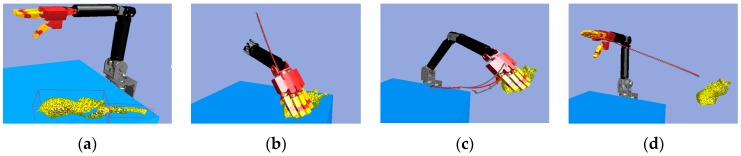
A procedure of grasping the fifth object in the simulator: (**a**) The target object (low stiffness category); (**b**) Collision-free paths for grasping; (**c**) RRT-Connect algorithm; (**d**) Successfully separated and placed in different boxes.

**Figure 50 sensors-19-01595-f050:**
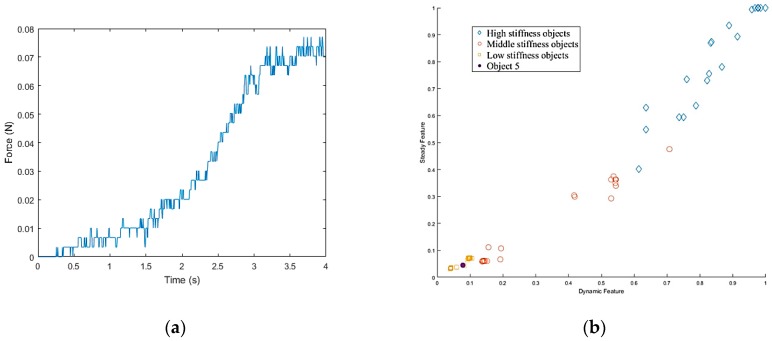
Object 5 stiffness measurement: (**a**) Stiffness; (**b**) K-NN.

**Figure 51 sensors-19-01595-f051:**
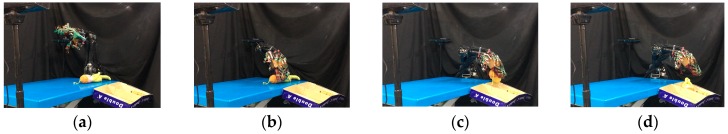
A procedure of grasping the fifth object: (**a**) The target object (low stiffness category); (**b**) Collision-free paths for grasping; (**c**) RRT-Connect algorithm; (**d**) Successfully separated and placed in different boxes.

**Figure 52 sensors-19-01595-f052:**
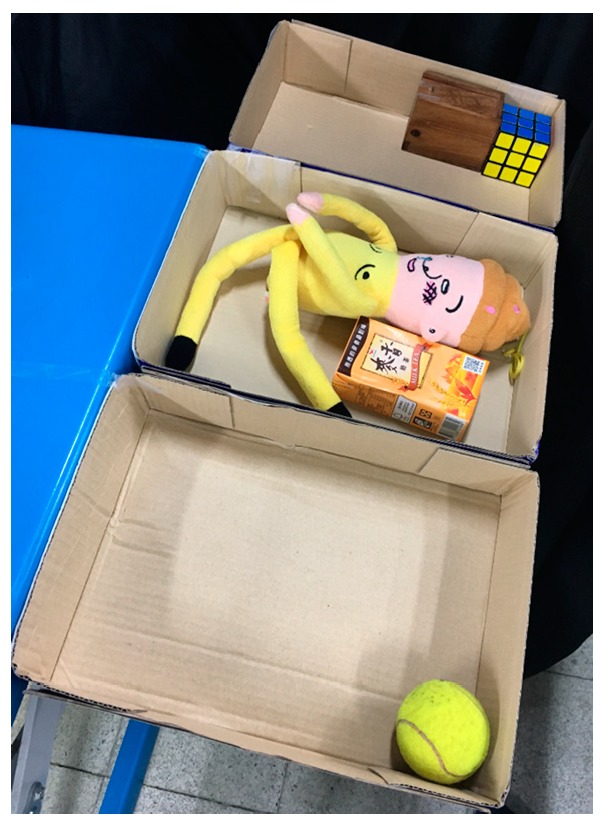
Classification results.

**Table 1 sensors-19-01595-t001:** The rules for inference of object logic states.

Pre-States	Actions	Current States
$s_{1}=\left\{0,0,0\right\}$	PickUp(object 1)	$s_{1}=\left\{1,0,-1\right\}$
$s_{1}=\left\{0,0,0\right\}$	PutDown(object 2, object 1)	$s_{1}=\left\{0,2,0\right\}$
$s_{1}=\left\{1,0,-1\right\}$	PutDown(object 1, object 2)	$s_{1}=\left\{0,0,2\right\}$

**Table 2 sensors-19-01595-t002:** The rules for segmentation point sets.

Action	Previous State	Current State	Detect Point Cloud in *Previous* Position		Detect Point Cloud in *Current* Position	
			Yes	No	Yes	No
PickUp(object 1)	$s_{1}=\left\{0,0,0\right\}$	$s_{1}=\left\{1,0,-1\right\}$	covered object	normal	object 1	err
PickUp(object 1)	$s_{1}=\left\{0,0,2\right\}$	$s_{1}=\left\{1,0,-1\right\}$	object 2	err	object 1	err
PutDown(ground)	$s_{1}=\left\{1,0,-1\right\}$	$s_{1}=\left\{0,0,0\right\}$	err	normal	object 1	err
PutDown(object 1, object 2)	$s_{1}=\left\{1,0,-1\right\}$	$s_{1}=\left\{0,0,2\right\}$	err	normal	object 1 and 2	err

**Table 3 sensors-19-01595-t003:** Multi-objective RRT-connect algorithm.

Input *k* = 1, *r* = 1, *m*, {$q_{init}$} and {$q_{goal1},\ldots ,q_{goaln}$}; $q_{rand}\leftarrow randomConfig()$; Find the nearest neighbor $n_{init}$ of $q_{rand}$ to $T_{init}$; Define *k*; Make a segment toward $q_{rand}$ from $n_{init}$ with a fixed incremental distance *d*; Define $q_{newinit}$; If the segment connecting $n_{init}$ and $q_{newinit}$ is collision free then extend $T_{init}$ (generate by {$q_{newinit}$, $T_{init}$}); Find the nearest neighbor $n_{goal}$ of $q_{rand}$ to $T_{goal}$; Make a segment toward $q_{rand}$ from $n_{goal}$ with fixed incremental distance *d*; Define $q_{new\mathrm{goal}}$; If the segment connecting $n_{goal}$ and $q_{new\mathrm{goal}}$ is collision free then extend $T_{goal}$ (generate by {$q_{new\mathrm{goal}}$, $T_{goal}$}); If $T_{init}$ and $T_{goal}$ are connected then Output the path, and end. Else replace *k* by *k* + 1, and return to Step 4 Else if *r* < *m* then replace *r* by *r* + 1, and return to Step 2 Else end; Else if *r* < *m* then replace *r* by *r* + 1, and return to Step 2 Else end;

**Table 4 sensors-19-01595-t004:** Sensor equipment of one finger.

Sensor Type	Count/Finger
joint torque {current (torque) control loop in BLDC motor}	3
joint position	3
motor speed	3
distributed tactile sensor arrays (376 detecting points) (Tekscan, Inc., South Boston, MA, USA)	1
three-axis force/torque (each fingertip)	1
six-axis force/torque (the wrist joint) (Mini 40, ATI Industrial Automation, Apex, NC, USA)	1

**Table 5 sensors-19-01595-t005:** Specification of NTU 6-DOF robot arm.

Total Weight	About 5.2 kg (exclude the shoulder)
Max. Payload	Over 2 kg
Max. Joint Speed	Exceed 110 deg./sec.
Max. Reachable distance	510 mm
Max. width	110 mm
Motor	Brushed DC X6
Reduction device	Harmonic drive
Transmission	Timing Belt, Gear

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
