# Peer review of "Robot Intelligent Grasp of Unknown Objects Based on Multi-Sensor Information"

_sensors, 2019, doi:10.3390/s19071595_

Round 1

Reviewer 1 Report

The robot hand is important end effector for robots to interact with the environment, this paper combined vision and robot hand real-time grasp control action to achieve reliable and accurate object grasping in a cluttered scene. The topic is interesting and the research has much application potential. But several aspects should be improved.

1. Some expression in the paper should be improved, such as “All experiments were conducted on the self-fabricated 6-DOF NTU robot arm and 21-DOF and 5-finger NTU hand. “in the abstract part.

2. Actually, the sentence:” Humans naturally grasp contact points that are always on the longer side of an object” is not correct in some situations.

3. The author said the intelligent real-time grasp system was achieved 16 that is reliable enough to handle various objects with unknown weights, frictions, and stiffness, but there is nothing about weight estimation in the paper, how the system can achieve the above goal?

4. Some figures are not clear.

Author Response

Article Revised Manuscript ID: sensors-467098

Article Title: Robot Intelligent Grasp of Unknown Objects Based on Multi-Sensor Information

To: sensors Editor

Re: Response to reviewers

Dear Editors and Reviewers,

Thank you for allowing resubmission of our manuscript, with an opportunity to address the reviewers’ comments.

Thank you very much for your careful review and constructive suggestions with regard to our manuscript “Robot Intelligent Grasp of Unknown Objects Based on Multi-Sensor Information” (Manuscript ID: sensors-467098).

Those comments are helpful for authors to revise and improve the paper. We have studied comments carefully and tried our best to revise and improve the manuscript and made some changes in the manuscript according to the referees’ comments.

A revised portion is marked in RED in the paper (our point-by-point response to the comments). Authors hope that the resubmitted manuscript will be reconsidered. Your assistance will be greatly appreciated. Please feel free to contact us with any questions and we are looking forward to your consideration. The main corrections in the paper and the responses to the reviewers’ comments are given below (pdf).

Reviewer 2 Report

The authors presented a well-written manuscript on the algorithm for robotic grasping using information derived from cameras, force/torque sensor, and tactile sensor arrays. By demonstrating the grasping of three distinctly different objects of different sizes and stiffnesses, they concluded robotic grasping could be performed without slipping. I have minor suggestions for them.

1) In Figure 21, it seems there is a sudden initial spike of force before decreasing, this shows there is a delay, if the object is fragile, it may be broken. It would be nice to reduce this possibility.

2) In the case of slip detection, rate of loading is also an important parameter. What would be the maximum rate of loading before the robot arm fails to grasp the object?

Author Response

(The authors gave the same response as above.)

Reviewer 3 Report

This paper is not presented in the format of a journal paper. 

The Introduction is just a repeat of the abstract and lacks the problem’s context and the related state of the art. Next, all sections are presented in subsections containing brief descriptions of concepts and not in a journal narrative-type style. 

It seems like a thesis or work report was just put into MDPI’s format and submitted for review. We can therefore explain its length (27 pages).

For this submission, I recommend its rejection. I recommend authors to extract the main concepts and contributions of their work and write an appropriate journal-type paper.

Author Response

(The authors gave the same response as above.)

Round 2

Reviewer 1 Report

Some figures ares still low in resolution and hard to read.

Author Response

(The authors gave the same response as above.)

Reviewer 3 Report

I do not see how the authors addressed my comment on the paper’s structure. I still believe that this work does not have a traditional journal paper structure and that it seems that a research report was just cut and pasted into MDPI’s format. We can see from the way it is written: a narrative describing the steps performed. 

This fact does not seem to be highlighted by the other two colleague reviewers. I will not insist on this matter anymore but I do have the following remarks:

1) There is no state of the art on the main topics addressed by the paper. The paper lacks a review of the most relevant grasping and path planning methods. This is necessary to better appreciate their limitations and the innovation of the research proposed in the paper.

2) Please remove the gray background of Matlab figures: 25, 26, 29, 30, 31.

3) Figure 33 has too many subfigures. Only the essentials must be shown (Is there a noticeable difference between figures a-c, d-f, and g-h?) Showing figures a, d, h, and i should be enough.

4) Section 5.2 Experiment 2 must be improved:

4.1) Figures 36, 39, 42, 45, 48, 51 are not easy discernable because both robot arm and objects are placed so far.

4.2) Plots comparing object stiffness will be more meaningful if they are all integrated in a single plot.

Author Response

(The authors gave the same response as above.)
